# Autoformalization of English-Described Mathematical Text

## Abstract

Autoformalization aims to convert natural language text into machine-verifiable formalizations, offering significant potential for advancing mathematical automation. However, the lack of high-quality parallel data between natural language and formal language in mathematics severely hinders the development of autoformalization. This paper proposes a linguistic-based method which converts English-described mathematical text into formal formulas. This method can automatically construct large-scale, high-quality formulas without model training. To evaluate the method, an experiment on the dataset including theorems and examples from *Calculus* is conducted, and the autoformalization result shows the method achieves an accuracy of 77.90%. And a case study of automatic verification of English-described mathematical proof is provided to demonstrate the usability of our method.

## 1 Introduction

Mathematical knowledge can be represented in both natural languages and formal languages. Natural languages provide flexibility and intuitiveness, and many mathematics-related researchers usually use English or their mother languages to describe their mathematical knowledge. As a comparison, formal languages, e.g. HOL Light (Harrison, 1996), Coq (Coq, 1996), Isabelle (Paulson, 1994), and Lean (de Moura & Ullrich, 2021), provide significant advantages in mathematical automation, including the resolution of geometric problems (Lu et al., 2021; Seo et al., 2015) and word problems (Huang et al., 2017; Mandal & Naskar, 2021). With the development of automatic theorem proving and AI for mathematics, they are becoming more important.

However, writing mathematical knowledge in one of these formal languages requires not only the knowledge itself but also expertise in the language. It is difficult for both area researchers who are accustomed to writing in natural languages and formal language programmers who have limited knowledge about the corresponding area. This difficulty highlights the importance of autoformalization, which converts natural language-described mathematical text into corresponding formal representations.

To address the challenge of autoformalization, researchers explore various approaches. In the rule-based methods, a controlled natural language is always proposed as the semi-formal language for mathematics description, and the autoformalization converts the controlled natural language described mathematical text into a logical form. Kühlwein et al. (2009) propose the Naproche System that enables the controlled natural language CNL to be parsed into first-order formulas. Schaefer & Kohlhase (2020) propose the Grammatical Logical Inference Framework, by which a controlled natural language system can be realized by syntax parsing, semantics construction and inference rules. Vershinin & Paskevich (2000) describe the language ForTheL to represent formal theories in mathematics and ForTheL-phrases can be mapped to first-order formulas by syntactical transformation. Pathak (2024) further introduces Simplified ForTheL, an extension of ForTheL, and implements its conversion to the Lean system. Besides these rule-based methods, Wang et al. (2018; 2020) convert English-described mathematical text into Mizar language by neural machine translation. Singh et al. (2020) and Lu et al. (2022) involve extracting logical propositions from natural language-described text by neural networks. However, the scarcity of parallel data between the natural language-described texts and the corresponding formal descriptions limits the progress of neural network-based approaches. Researchers tend to circumvent this challenge by manually cu-

rating small corpora or by using few-shot learning with LLMs (Wu et al., 2022; Gadgil et al., 2022; Agrawal et al., 2022). Besides, Jiang et al. (2023; 2022), Gao et al. (2024), and Wang et al. (2024) also use LLMs to convert formal descriptions into natural language-described texts for dataset construction. However, since most formal languages are unable to fully capture the semantics of a natural language, the natural language text produced by these approaches lacks the inherent diversity and flexibility in expression, which is rigid and unlike natural language. If the dataset is used to train neural network-based autoformalization models, the robustness and generalization of the model could be weakened.

To address the scarcity of high-quality parallel data between natural language-described texts and the corresponding formal descriptions, this paper proposes a linguistics-based method for the autoformalization from English-described mathematical text to predicate logic-based formulas. These predicate logic-based formulas not only preserve the semantic information of a sentence but also maintain the logical structure of the text. To show the effectiveness of this method, experiments are conducted on sentences of theorems and examples from the 9th edition of *Calculus* by James Stewart.

## 2 PREDICATE LOGIC-BASED FORMULA

Currently, commonly used formal languages, e.g., Isabelle, Coq, and Lean, are designed for different purposes but exhibit certain semantic information loss compared to natural language. As other information, such as aspect, modality and kinds of adverbials may still be useful for further analysis or text inference, a formal representation that maintains all information of English-described mathematical text would be appreciated. Therefore, a representation based on predicate logic is proposed for formalizing English-described mathematical text in this section.

### 2.1 PREDICATE

The core logical structure of an English-described mathematical text lies in the predicate relations between mathematical objects and the logical relations between predicates. To represent a predicate relation, an atomic formula $P$ is defined as the following form:

$$P ::= V_{node}(N_{node})^+$$

Here, $P$ is also called a predicate and is defined by a verb related component $V_{node}$ and a set of noun related components $N_{node}$. The symbol $+$ denotes the presence of one or more subcomponents.

Compared with the predicate logic, an English-described mathematical text may also contain several modifications. For example, in "A function $f$ is called increasing on an interval $I$ if $f(x_1) < f(x_2)$ whenever $x_1 < x_2$ in $I$", the prepositional phrase "on an interval $I$" modifies the predicate verb "is called", specifying the interval under which the function $f$ is considered increasing. In "The function $f$ never takes on the same value twice", the adverbial "never" and "twice" modify the predicate verb "takes" from the perspectives of negation and frequency respectively. If only the verb is kept to represent the predicate relations, the semantic information will be lost. Thus, $V_{node}$ should be characterized by the core verb and the corresponding auxiliary information. The auxiliary information not only expresses modifier relations and aspect but also alters the validity of predicate relations. Then the form of a $V_{node}$ is defined as follows:

$$V_{node} ::= V < POS_V > [AV < POS_{AV} >]^* [ADV]^*$$

where $V$ denotes the base form of the core verb, $AV$ denotes an auxiliary verb, $< POS_V >$ denotes aspect and other part of speech information of the core verb, $< POS_{AV} >$ denotes aspect and other part of speech information of the auxiliary verb, $ADV$ denotes an adverbial, and the symbol $*$ denotes the zero or more presence of corresponding components. Similarly, $N_{node}$ is also characterized by the core noun and the corresponding auxiliary information, i.e.:

$$N_{node} ::= N [ADJ]^*$$

where $N$ denotes the core noun, and $ADJ$ denotes the auxiliary information or relevant attributes of the core noun.

## 2.2 Predicate Logic-based Formula

Predicates represent the basic relations between mathematical entities, while the representation of mathematical concepts and relations among complex entities should also involve connective symbols. Besides, there are also temporal conjunctions, contrastive conjunctions, and other conjunctions or guiding words in English-described mathematical text. Therefore, it is necessary to distinguish these conjunctive relations between predicates in formal representation. In this paper, $\circ$ and $\square$ are introduced to represent temporal and contrastive conjunctions respectively; furthermore, if an undefined conjunction or guiding word is occurs in a text, the capitalized form of the conjunction or guiding word will be used to represent the corresponding relation in its formal representation. Then, a predicate logic-based formula can be recursively defined as follows:

1. A predicate $P$ is an atomic formula.
2. If $A$ is a formula, then so is $\neg A$.
3. If $A$ and $B$ are formulas, then so $A \wedge B$, $A \vee B$, $A \rightarrow B$, $A \leftrightarrow B$, $A \circ B$, $A \square B$, and A $cap$ B are all formulas.
4. If $A$ is a formula, then so are $V_{node}(A)^{+}$ and $V_{node}(N[A]^{*})^{+}$.

In this definition, atomic formulas represent affirmative simple sentences and a negation symbol in Rule 2 maps negative sentences into formulas. Rule 3 and Rule 4 formalize complex sentences in English-described mathematical text: Rule 3 constructs formulas using connective symbols, to map compound sentences and adverbial clauses into formal formulas, where "$cap$" is a uniform representation for any undefined relation or guiding word; Rule 4 employs recursive structures of formulas to map noun clauses, predicative clauses, and adjective clauses. These four rules establish the map between the syntax structures of English sentences and the corresponding formula representation.

Based on this definition, the automatic conversion from English-described mathematical text to predicated logic-based formulas can be constructed and it is also not difficult to convert these predicated logic-based formulas into commonly used formal languages by eliminating unnecessary information.

# 3 Generation of Predicate Logic-based Formula

The framework of the linguistic-based autoformalization is shown in Figure 1. An English-described mathematical text is parsed by the Stanford CoreNLP tool and LaTeX parser: the Stanford CoreNLP tool parses a sentence into a syntactic parse tree, and the LaTeX parser converts a mathematical formula into a formula tree which then is integrated with the syntactic parse tree. Clause analysis and sentence component analysis are performed on a syntactic parse tree to construct a standardized grammar tree with clause and sentence component information. Lastly, a predicate logic-based formula is obtained by using mapping rules between standardized grammar trees and predicate logic-based formulas.

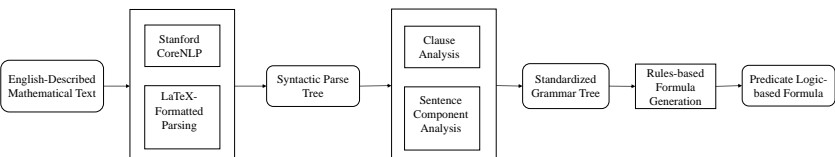

Figure 1: Framework of math text autoformalization

## 3.1 Syntactic Parse Tree

To obtain the structure and components of an English-described mathematical text, it is necessary to do part-of-speech tagging, syntactical structure analysis, and LaTeX formula analysis. In this paper, the Syntactic Parse Tree (SPT) that originated from generative grammar theories is employed to represent the hierarchical syntactic structure of English-described mathematical texts. Each node of SPT corresponds to a grammatical component, explicitly capturing the syntactic relationships between words and phrases. For instance, in the sentence "If the left and right limits are not equal, then the limit does not exist", SPT (as shown in Figure 2) consists of clause node $SBAR$, adverb phrase

node $ADVP$, noun phrase node $NP$, and verb phrase node $VP$, clearly illustrating how grammatical elements organize to form meaningful sentences. These abbreviations (e.g., VBZ, VBD, NP, SBAR) are standardized notations for grammatical categories and phrase structure labels in computational linguistics. Their specific definitions can be referenced in the official Penn Treebank documentation (Taylor et al., 2003). In this paper, Stanford CoreNLP is used to generate the general SPT from an English text, and it uses a neural network-based approach that not only accelerates the parsing process but also improves accuracy in syntactic parsing.

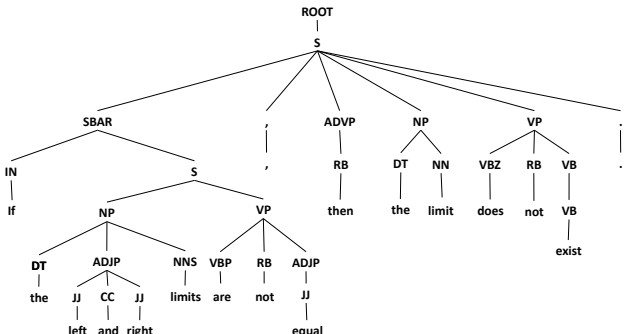

Figure 2: Syntactic parse tree

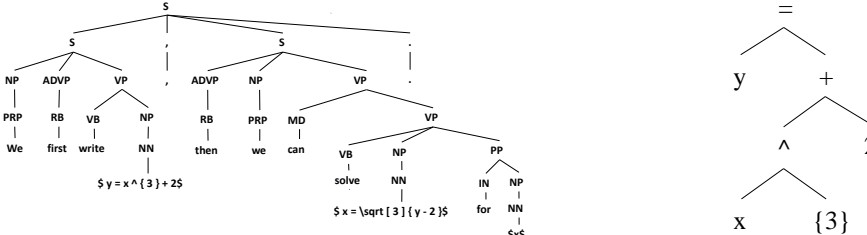

Figure 3: Syntactic parse tree with formula       Figure 4: Syntax tree of math formula

However, there are also LaTex-formatted mathematical symbols or formulas which can not be handled by the Stanford CoreNLP, and a direct application of Stanford CoreNLP on them gives an error message. Therefore, a LaTeX parser is required to generate syntactic parse trees for math formulas formatted in LaTeX. LaTeX content enclosed by $ is first replaced with placeholders, enabling syntactic analysis of the remaining pure English text. If a predicate verb already exists in the relevant clause, the LaTeX-formatted content is labeled as a noun phrase; otherwise, it is labeled as a clause. Then each LaTeX-formatted formula is parsed by a LaTeX parser, and the syntax tree of the math formula will be inserted into the corresponding place of the general syntactic parse tree. Take "We first write $y = x^3 + 2$, then we can solve $x = \sqrt[3]{y-2}$ for $x$" as an example, the formulas $y = x^3 + 2$, $x = \sqrt[3]{y-2}$ and $x$ are labeled with $NN$ first (see in Figure 3), and the syntax tree of $y = x^3 + 2$ (see in Figure 4) will be placed in the corresponding position of $y = x^3 + 2$ in Figure 3. The syntax trees of the math formulas will be used for further symbolic calculation or step-by-step verification.

### 3.2 STANDARDIZED GRAMMAR TREE

Generative grammar primarily focuses on defining the syntactic structure of sentences, such as phrase hierarchies and grammatical rules, but overlooks entities and logical relationship extraction. However, this information is essential for generating predicate logic-based formulas, which are necessary for inference and verification. To address this limitation, we turn to systemic functional grammar (SFG), a framework developed by M.A.K. Halliday. SFG emphasizes the connection between syntactic forms and their communicative and semantic functions, allowing for a clear distinction between "entities", "relations", and "modifiers". Building on SFG, we propose the standardized grammar tree (SGT). The SGT is not only intuitive but also effectively bridges the gap between syntactic parse trees and predicate logic-based formulas, enabling a smooth conversion for formalization.

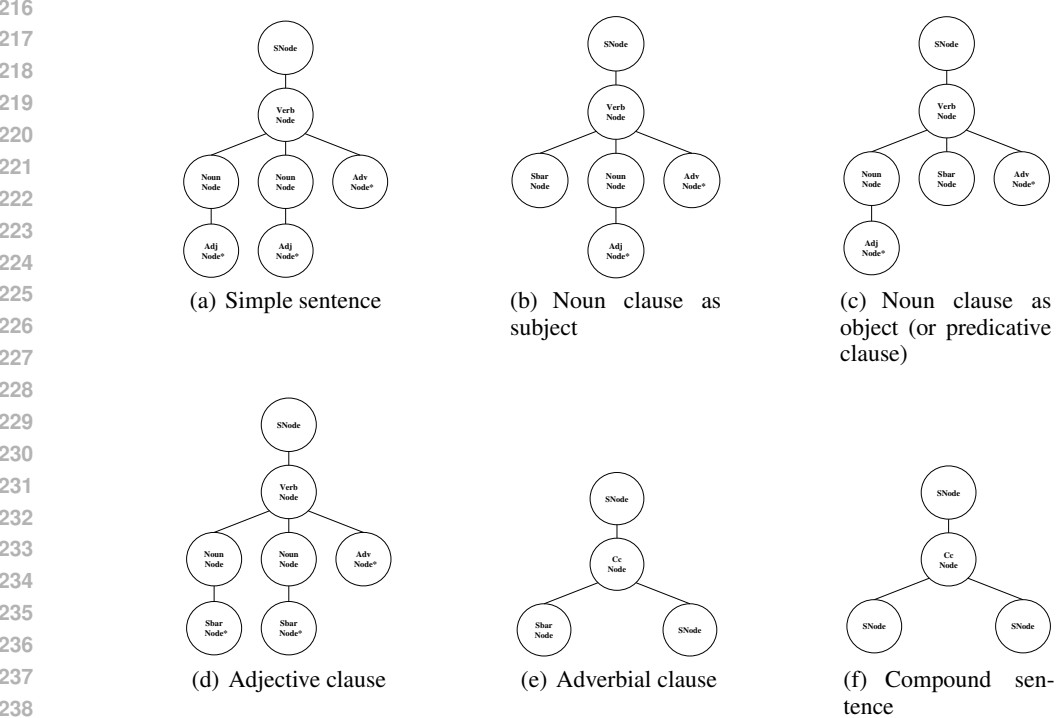

(a) Simple sentence

(b) Noun clause as subject

(c) Noun clause as object (or predicative clause)

(d) Adjective clause

(e) Adverbial clause

(f) Compound sentence

Figure 5: Standardized grammar tree

A SGT is constructed from some elementary nodes which include $VerbNode$, $NounNode$, $CcNode$, $AdjNode$, $AdvNode$, $SNode$, and $SbarNode$: $VerbNode$ records all verb information of a predicate, which may include core verb, modal verb, and auxiliary verbs; $NounNode$ records all information of a subject or an object; $CcNode$ records the guide word or conjunction; $AdjNode$ records the word or phrase that modify $NounNode$; $AdvNode$ records the word or phrase that modify $VerbNode$; $SNode$ and $SbarNode$ records the type of a sentence, i.e. a simple one or a complex one. These nodes can correspond to "entities", "relations", and "modifiers" in the SFG. According to our analysis, the commonly use grammar structures of English sentences are summarized in Figure 5: an argument relation between a $VerbNode$ and a $NounNode$ is represented by a parent-child structure in SGT, then all arguments ($NounNode$) of a predicate are the direct children of the corresponding $VerbNode$. Similarly, a modification relation between a $AdjNode$ and a $NounNode$, or between a $AdvNode$ and a $VerbNode$, is also represented by a parent-child structure.

For complex sentences, standardized grammar trees can be constructed by adjusting simple sentences: In the SGT of a noun clause or a predicative clause, one argument child of the $VerbNode$ is not a $NounNode$ but a $SbarNode$ instead. In the SGT of an adjective clause, the $SbarNode$ is set as a adjective child of a $NounNode$. For adverbial clauses and compound sentences, the guide word or the conjunction is recorded in $CcNode$: in an adverbial clause, a subordinate clause is connected to the main clause by a guide word; in a compound sentence, clauses are connected together by a conjunction. Take "If the left and right limits are not equal, then the limit does not exist" as an example, the SPT and SGT are shown in Figure 6. As more detailed part-of-speech information is stored in the deeper node of SGT, a more concise structure representation is obtained: the predicate relation and the logical relation are clearly represent, which make automatic reasoning and verification easier.

To sum up, a standardized grammar tree is inductively defined as: 1) A structure with $SNode$ as the root and consisting of $VerbNode$, $NounNode$, $AdjNode$, and $AdvNode$, as the example in Figure 5(a), constitutes a standardized grammar tree; 2) If $T$ is a standardized grammar tree, the tree formed by replacing any $NounNode$ in $T$ with $SbarNode$ remains a standardized grammar tree; 3) If $T$ is a standardized grammar tree, the tree formed by replacing any $AdjNode$ in $T$ with $SbarNode$ remains a standardized grammar tree; 4) If $T_1$ and $T_2$ are both standardized grammar trees, the tree

formed by connecting $T_1$ and $T_2$ with $CcNode$ as the parent node remains a standardized grammar tree.

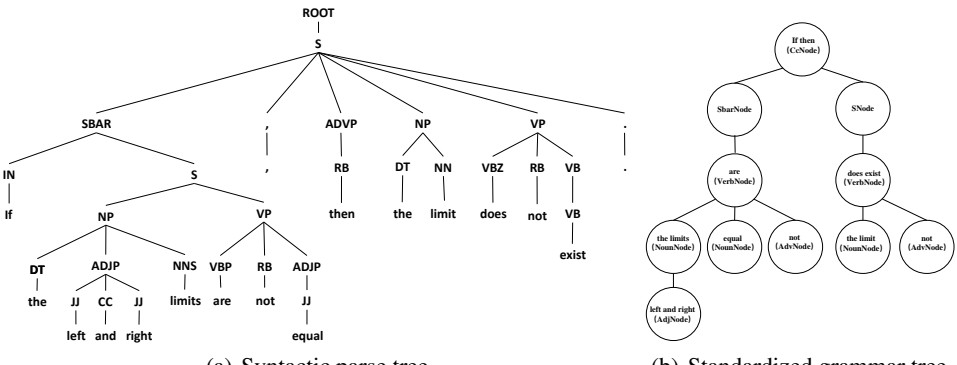

(a) Syntactic parse tree          (b) Standardized grammar tree

Figure 6: Comparison between SPT and SGT

### 3.3 CONVERSION FROM SPT TO SGT

In order to construct the standardized grammar tree from a given syntactic parse tree, clause analysis and sentence component analysis need to be performed.

Clause analysis aims to identify the type of grammar structure of a sentence. This can be accomplished by analyzing the syntactic structure features of the sentence, which are exhibited by the structural relations the $SBAR$ node and its associated $S$, $VP$, $NP$ or $CC$ nodes. The typical syntactic structure features of different types of sentences are shown in Figure 7.

1. The typical feature of a noun clause as a subject is the SPT whose $S$ node has a $SBAR$ node and a $VP$ node as its children and the $SBAR$ is just on the left of the $VP$.
2. The typical feature of a noun clause as an object is the SPT whose $VP$ node has a node of the $VB$ family (e.g. $VBZ$, $VBP$, etc.) node and a $SBAR$ node as its children and the $SBAR$ is just on the right of the node of the $VB$ family.
3. The typical feature of a predicative clause is similar to a noun clause as an object, but they can be distinguished based on the type of the core verb. A predicative clause typically follows a linking verb, whereas a noun clause as an object is generally found after a notional verb.
4. The typical feature of an adjective clause is the SPT whose $NP$ node has a $NP$ node and a $SBAR$ node as its children and the $SBAR$ is just on the right of the $NP$.
5. The typical feature of a compound sentence is the SPT whose $S$ node has a $CC$ node and two $S/SBAR$ nodes as its children.
6. There are a set of structure features for adverbial clauses, and a general identify method is to show that a $SBAR$ comes before or after a combination of an $NP$ and a $VP$, e.g. Figure 7 (e-f). Besides, some special guide words can also be used to identify adverbial clauses, e.g., "if", "unless", "although", etc.

After clause analysis, the sentence type and the possible conjunction or guide word can be obtained. Besides, the clause analysis will be recursively performed on the first SBAR subtree, and the rest of the syntactic parse tree is usually the main clause or a coordinate clause. The main clause or the coordinate clause will be reanalyzed to determine if there exists more clauses. For complex sentences with multiple clauses, the clause analysis is performed recursively.

Sentence component analysis aims to identify various components within a simple sentence or clause. This can be accomplished by analyzing syntactic structure features and part-of-speech tags.

The syntactic structure features of sentence components are listed as follows: A subject is usually recorded in the first $NP$ child of the $S$ node. A predicate is usually located under the first $VP$ child of $S$ node, but for sentences containing auxiliary verbs, multiple rounds of searching need to be performed under VP to find the core verb: if the core verb is a notional verb, objects are stored in the followed $NP$s; if the core verb is a linking verb, predicative is stored in the followed $NP$

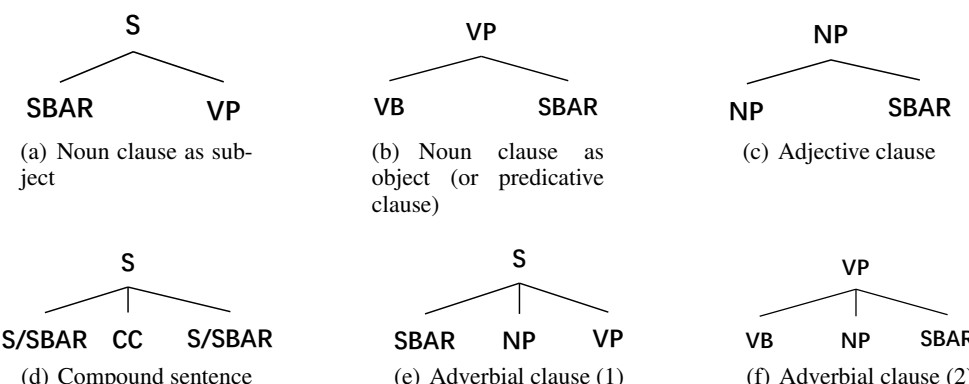

Figure 7: Typical syntactic structures of clause analysis

or $ADJP$. In a sentence, $JJ$, $RB$, $ADJP$, $ADVP$, or $PP$ that followed a modified word can be classified as an adjective or adverbial based on the corresponding part-of-speech tagging.

The generation of a standardized grammar tree is achieved though the recursive algorithm GenSGT (as shown in the appendix) : This algorithm takes a SPT as input and outputs the corresponding SGT. If the sentence with respect to the input SPT is a simple sentence, component analysis is done to obtain the corresponding SGT nodes, and the SGT can be constructed accordingly; otherwise, the SPT of the first obtained clause and the rest part of the SPT are processed by GenSGT recursively; furthermore, if the sentence type is an adverbial clause or a compound sentence, a $CcNode$ is constructed. Finally, the SGT is constructed based on the sentence type, subSGTs and CcNode.

### 3.4 FORMULA GENERATION

The standardized grammar tree can be converted into a predicate logic-based formula by using mapping rules. For a simple sentence, information of $V < POS_V >$ and $AV < POS_{AV} >$ in $V_{node}$ can be obtained from $VerbNode$, information of $ADV$ can be obtained from the $VerbNode$'s child $AdvNode$; similarly, the core noun $N$ in $N_{node}$ is obtained from $NounNode$, and the corresponding $ADJ$ can be obtained from $NounNode$'s child $AdjNode$; for complex sentences, the corresponding formulas can be constructed by rules provided Table 1, and the mapping rules for commonly used conjunctions or guide words are presented in Table 2.

With the above mapping rules, a formula that contains all semantic information of the original sentence can be constructed from a SGT. Taking the SGP in Figure 6(b) as an example, its corresponding formula is "$\{\neg be<VBP>(the\ limits[left \wedge right], equal)\} \rightarrow \{\neg exist<VB>[do<VBZ>](the\ limit)\}$."

Table 1: Mapping rules for different types of complex sentences

| Sentence Type | Predicate Logic-based Formulas Structure |
|---|---|
| negative sentence | $\neg P$ |
| noun clause as subject | $V_{node}(P, N_{node})$ |
| noun clause as object (or predicative clause) | $V_{node}(N_{node}, P)$ |
| adjective clause | $V_{node}(N[P], N_{node})$ |
| adverbial clause | $P_1 \rightarrow P_2$, $P_1 \leftrightarrow P_2$, $P_1 \circ P_2$, etc. |
| compound sentence | $P_1 \wedge P_2$, $P_1 \vee P_2$ |

## 4 EXPERIMENTS

### 4.1 CORRECTNESS EVALUATION

To verify the effectiveness of the formalization method, a dataset is constructed by extracting sentences of theorems and examples from the 9th edition of *Calculus* by James Stewart. The dataset

Table 2: Mapping rules for commonly used conjunctions or guide words

| Connective Symbols | Conjunctions or Guide Words |
|---|---|
| $\wedge$ | and, both and, not only but also, etc. |
| $\vee$ | or, either or, etc. |
| $\rightarrow$ | if, because, since, as, so, etc. |
| $\leftrightarrow$ | if and only if, likewise, etc. |
| $\circ$ | then, after, etc. |
| $\square$ | but, although, whereas, etc. |
| $cap$ | undefined relations |

consists of 4,339 sentences, including both simple and complex sentences. Each sentence was processed by the linguistic-based method proposed in this paper, and the correctness of generated formulas were evaluated: A formula is correctly converted if the form of the formula is correct and all clauses and sentence components information is successfully extracted.

Table 3: Autoformalization result analysis

| | |
|---|---|
| Correct | 3380 (77.90%) |
| Errors in part-of-Speech Tagging | 127 (2.93%) |
| Errors in Parse Tree Generation | 259 (5.97%) |
| Incomplete structure feature collection | 573 (13.20%) |

The result in Table 3 shows that 77.90% of the sentences were successfully converted into the predicate logic-based formulas and a detailed analysis of the incorrect conversion is summarized as follows:

1. Errors in part-of-speech tagging: As Stanford CoreNPL tags part-of-speech by a neural network-based approach, errors of misclassify may encounter. Tagging verbs as nouns is a common mistake, e.g. "changes" was tagged as a noun in "If $f'$ changes from positive to negative at $c$, then $f$ has a local maximum at $c$" The incorrect tagging will mislead sentence component analysis and predicate extraction.

2. Errors in parse tree generation: Stanford CoreNLP may also exhibit inaccuracies when processing sentences with complex structures and it is particularly evident in sentences with conjunctions "and" and "or". For example, in "If $f'(x) > 0$ for all $x < c$ and $f'(x) < 0$ for all $x > c$, then $f(c)$ is the absolute maximum value of $f$", the conjunction "and" is intended to link two complete clauses. However, Stanford CoreNLP erroneously treated it as the conjunction that joints two mathematical expressions, which led to an incorrect syntactic structure analysis and ultimately resulted in an inaccurate formalization.

3. Incomplete structure feature collection: Incomplete syntactic structure collection is also one cause for incorrect conversion. The structure features for elliptical sentences have not been defined, which causes the incorrect syntactic structure feature analysis from SPT to SGT. For example, in "The conic is an ellipse if $e < 1$, a parabola if $e = 1$, or a hyperbola if $e > 1$", the second and third clauses omit both the subject and the copula, which leads to incorrect clause analysis and sentence component analysis, and fails to generate correct formulas.

It should be noted that even formulas that are not correctly converted can be manually corrected with ease. Furthermore, syntactic structure feathers of more types of sentences can be added to formalize more kinds of mathematical sentences. The codes of the linguistic-based autoformalization method and the dataset can be found on the Gitee repository of the project: `https://gitee.com/auto-formalization/logicalsemanticanalysis.git`.

## 4.2 CASE STUDY: AUTOMATIC VERIFICATION OF MATHEMATICAL PROOF

To show the practicality of the linguistic-based method for autoformalization, this paper proposes an automated verification framework for mathematical proofs. The framework takes English-described mathematical text that includes theorems, premises, reasoning steps, and conclusion as input, converts these texts into predicate logic-based formulas, and automatically verifies the the correctness of

reasoning steps and conclusion by logical reasoning methods and the symbolic computation library SymPy (2025).

Take one theorem from Calculus as an example: the text is divided into the pre-proved theorems, premises, reasoning steps and conclusion (see in Table 4), the linguistic-based autoformalization approach automatically generates the predicate logic-based formulas (see in Table 5), and the results of step-by-step verification is shown in Table 6 where the correctness of all reasoning steps and the conclusion are verified to be correct. It should be noted that some of the auxiliary verbs and part-of-speech tags in the example are unnecessary for the automatic verification, thus the predicate logic-based formulas can be simplified by removing them.

Table 4: Representation of theorems, premises, reasoning steps and conclusion in English

| Theorems: | If $\lim_{x \to a} f(x) \neq f(a)$, $f$ is not continuous at $a$. |
|---|---|
| Premises: | If $x = 2$, $f(x) = 1$. |
| | If $x \neq 2$, $f(x) = \frac{x^2 - x - 2}{x - 2}$. |
| Reasoning Steps: | Here $f(2) = 1$ is defined and $\lim_{x \to 2} f(x) = \lim_{x \to 2} \frac{x^2 - x - 2}{x - 2}$ |
| | $= \lim_{x \to 2} \frac{(x-2)(x+1)}{x-2} = \lim_{x \to 2}(x + 1) = 3$ exists. |
| | But $\lim_{x \to 2} f(x) \neq f(2)$, so $f$ is not continuous at 2. |
| Conclusion: | $f$ is not continuous at 2. |

Table 5: Formula representation of theorems, premises, reasoning steps and conclusion

| Theorems: | $\{\text{hold}(\lim_{x \to a} f(x) \neq f(a))\} \to \{\neg\text{be}(f, \text{continuous[at } a])\}$ |
|---|---|
| Premises: | $\{\text{hold}(x = 2)\} \to \{\text{hold}(f(x) = 1)\}$ |
| | $\{\text{hold}(x \neq 2)\} \to \{\text{hold}(f(x) = \frac{x^2 - x - 2}{x - 2})\}$ |
| Reasoning Steps: | $\{\text{define[here]}(f(2) = 1)\} \wedge \{\text{exist}(\lim_{x \to 2} f(x) = \lim_{x \to 2} \frac{x^2 - x - 2}{x - 2}$ |
| | $= \lim_{x \to 2} \frac{(x-2)(x+1)}{x-2} = \lim_{x \to 2}(x + 1) = 3)\}$ |
| | $\{\text{hold}(\lim_{x \to 2} f(x) \neq f(2))\} \to \{\neg\text{be(f, continuous[at } 2])\}$ |
| Conclusion: | $\neg\text{be}(f, \text{continuous[at } 2])$ |

Table 6: Step-by-step verification based on formal formulas

| Reasoning$_i$ | Required premises and theorems | Verification Method |
|---|---|---|
| define[here]($f(2) = 1$) | $\{\text{hold}(x = 2)\} \to \{\text{hold(f(x)=1)}\}$ | Symbolic Calculation |
| exist($\lim_{x \to 2} f(x) = \lim_{x \to 2} \frac{x^2 - x - 2}{x - 2}$ $= \lim_{x \to 2} \frac{(x-2)(x+1)}{x-2} = \lim_{x \to 2}(x + 1) = 3$) | $\{\text{hold}(x \neq 2)\} \to \{\text{hold}(f(x) = \frac{x^2 - x - 2}{x - 2})\}$ | Symbolic Calculation |
| hold($\lim_{x \to 2} f(x) \neq f(2)$) | $f(2) = 1, \lim_{x \to 2} f(x) = 3$ | Symbolic Calculation |
| $\neg\text{be}(f, \text{continuous[at } 2])$ | $\lim_{x \to 2} f(x) \neq f(2)$ $\{\text{hold}(\lim_{x \to a} f(x) \neq f(a))\} \to$ $\{\neg\text{be}(f, \text{continuous[at } a])\}$ | Logical Reasoning |

## 5 CONCLUSION

This paper proposes a linguistic-based method for autoformalization. The method can automatically convert English-described mathematical texts to predicate logic-based formulas without model training: an English-described mathematical sentence is converted into a syntactic parse tree by using the Stanford CoreNLP tool and a Latex parser; then clause analysis and sentence component analysis are conducted on the syntactic parse tree to obtain the corresponding standardized grammar tree; finally, a set of mapping rules is applied for formula generation. The predicate logic-based formulas not only preserve the semantic information of original sentences but also keep the logical structures of the mathematical text. An experiment on a dataset of 4,339 sentences shows that the linguistic-based automformalization achieves an accuracy of 77.90%, which can greatly reduce the work of manual formalization. The method can be used for the automatic verification of informal proofs and be beneficial for constructing high-quality parallel formula datasets for English-described text. Future work will focus on enlarging structure feature collection to improve the accuracy and using the method in AI for mathematics.

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

## A  APPENDIX

---

**Algorithm 1:** Generation of Standardized Grammar Tree(GenSGT)

---

**Input:** a SPT
**Output:** a SGT
clauseInfo ← ClauseAnalysis(SPT);
**if** *clauseInfo.type is a simple sentence* **then**
  componentInfo ← SentenceComponentAnalysis(SPT);
  SGTNodes ← ConstructSGTNodes(componentInfo);
  SGT ← ConstructSGT(SGTNode,clauseInfo.type);
  **return** SGT;
**else**
  $subSGT_1$ ← GenSGT(clauseInfo.getSubSPT);
  $subSGT_2$ ← GenSGT(clauseInfo.getRestSPT);
  **if** *clauseInfo.type is a adverbial clause or compound sentence* **then**
    conjOrGuide ← clauseInfo.getConjOrGuide;
    CcNode ← ConstructSGTNodes(conjOrGuide);
    SGT ← ConstructSGT($subSPT_1$,$subSPT_2$,CcNode,clauseInfo.type);
  **else**
    SGT ← ConstructSGT($subSPT_1$,$subSPT_2$,clauseInfo.type);
  **return** SGT;

---

