# OpenReview forum: "Autoformalization of English-Described  Mathematical Text"
_ICLR.cc/2026/Conference — Submitted to ICLR 2026_

### Official Review · Reviewer_PzaP · 2025-10-22

**Soundness:** 2
**Presentation:** 1
**Contribution:** 2
**Rating:** 2
**Confidence:** 4

**Summary:**

This paper proposes an autoformalization pipeline that translates English mathematical text into predicate logic–based formulas. The pipeline first employs Stanford CoreNLP and LaTeX-formatted parsing to generate a syntactic parse tree, then applies clause analysis and sentence component analysis through the recursive algorithm GenSGT to construct a standardized grammar tree. Finally, it derives logical formulas via predefined mapping rules. Experiments on calculus problems demonstrate that the proposed method achieves 77.9% accuracy, accompanied by detailed error analysis. The authors also present an illustrative example to verify the correctness of the generated predicate formulas.

**Strengths:**

* The paper presents an intuitive approach that leverages linguistic methods for autoformalization.
* The experiments demonstrate that the proposed method achieves reasonable accuracy on a newly constructed dataset.

**Weaknesses:**

* The paper is not well written and is difficult to follow, especially in Sections 3.2 and 3.3. The presentation is somewhat distracting, and I have several questions after reading — see below.
* The experimental evaluation appears rather limited and somewhat toyish, lacking comparisons with existing baselines. There are numerous recent autoformalization methods that use LLMS to formalize mathematical problems or theorems into Lean, Isabelle, or first-order logic. However, this paper does not include any comparison with such LLM-based approaches. Moreover, the statements used in the experiments seem relatively simple compared to those in existing benchmarks involving high-school competition or undergraduate-level mathematics. I strongly recommend evaluating on established autoformalization datasets and comparing with LLM-based baselines [1, 2, 3]. See also the recent survey [4] for a broader overview of related work.
* The paper only presents a single simple example to verify the generated formulas, without providing details of the verification procedure. It remains unclear whether the proposed method can generalize to more complex or diverse mathematical statements.

**References**
[1] *ProofNet: Autoformalizing and Formally Proving Undergraduate-Level Mathematics*
[2] *Autoformalize Mathematical Statements by Symbolic Equivalence and Semantic Consistency*
[3] *Rethinking and Improving Autoformalization: Towards a Faithful Metric and a Dependency Retrieval-based Approach*
[4] *Autoformalization in the Era of Large Language Models: A Survey*

**Questions:**

* I am a bit confused about the evaluation process. How does the paper ensure that the autoformalized statements are semantically aligned with the original natural language theorems? Was this verified through manual inspection or by some automated metric?
* It is also unclear what is meant by ''using symbolic calculation and logical reasoning as the verification method.'' Has this verification component been fully implemented? If so, how are mathematical operators such as limits, differentiation, and continuity handled — for instance, are these processed through SymPy?

---

### Official Review · Reviewer_BC3S · 2025-10-27

**Soundness:** 2
**Presentation:** 2
**Contribution:** 2
**Rating:** 4
**Confidence:** 2

**Summary:**

The paper proposes a linguistically grounded framework for automatically converting English mathematical descriptions into predicate logic–based formal expressions. Unlike most neural or purely rule-based approaches, this work introduces a hybrid method built upon syntactic and semantic analysis using Stanford CoreNLP and a LaTeX parser, followed by a structural normalization step that converts the syntactic parse tree (SPT) into a standardized grammar tree (SGT). Predicate logic formulas are then generated via explicit mapping rules. The authors also construct a dataset from *James Stewart’s Calculus (9th Edition)* and report an overall accuracy of 77.9% for autoformalization. Additionally, a case study demonstrates that the generated logical formulas can be used for automatic verification of mathematical proofs via symbolic computation (SymPy).

**Strengths:**

- The linguistic-based framework is very novel. The paper introduces a clear and interpretable rule-based pipeline that bridges natural and formal mathematical languages without any training data.
- The proposed technique is comprehensive. The inclusion of both syntactic and LaTeX parsing ensures the method handles mathematical notation and linguistic structure simultaneously.
- The code is open-sourced. The authors release their code and dataset publicly, which enhances the credibility and utility of the work.

**Weaknesses:**

- The scalability of the proposed approach may be limited, as it relies heavily on rule-based structures that might not generalize well to more diverse or less grammatically regular mathematical texts. Furthermore, its expressive capacity remains somewhat constrained: although predicate logic provides a powerful formalism, it can still fail to capture subtle aspects of mathematical semantics, such as implicit quantifiers, contextual assumptions, or omitted logical dependencies.
- The performance of the framework relies heavily on Stanford CoreNLP, whose output directly influences all subsequent stages of the pipeline. Even minor parsing inaccuracies can propagate through the system and compound into substantial downstream errors, ultimately affecting the overall accuracy of the autoformalization process.
- The evaluation dataset from a single textbook limits the generalizability of the results across mathematical subdomains or writing styles. Besides, the paper should be strengthened by benchmarking against recent LLM-based autoformalization approaches.
- It would be valuable to include alternative evaluation methods for assessing autoformalization correctness, such as Beq-based metrics [1, 2] or equivalence checking techniques [3, 4]. Incorporating these evaluations could provide deeper insight into the capabilities and limitations of the proposed approach.
- Some relevant recent work appears to be missing from the discussion [5].

   [1] Rethinking and Improving Autoformalization: Towards a Faithful Metric and a Dependency Retrieval-based Approach, ICLR 2025

   [2] Improving Autoformalization using Type Checking, arXiv 2025

   [3] Autoformalizing euclidean geometry, ICML 2024

   [4] Autoformalize Mathematical Statements by Symbolic Equivalence and Semantic Consistency, NeurIPS 2024

   [5] Autoformalization in the Era of Large Language Models: A Survey, arXiv 2025

**Questions:**

-  The proposed framework is evaluated on examples drawn exclusively from Calculus. How well do you expect the system to generalize to other mathematical domains, such as algebra, geometry, or discrete mathematics, where linguistic patterns and symbolic conventions differ significantly?
- Many mathematical texts contain implicit assumptions (e.g., "let $f$ be continuous") or ambiguous constructs that depend on context. How does the current framework handle such implicit semantics or context-dependent meaning, and could predicate logic be extended to represent them more faithfully?
- Is there a way to integrate the proposed method with LLM-based approaches (e.g., for parse correction or semantic disambiguation)? If so, how might such hybridization improve accuracy and scalability?
- The shared code is somewhat difficult to follow; it would be helpful to include a detailed README explaining how to run it.

---

### Official Review · Reviewer_NDCF · 2025-10-31

**Soundness:** 2
**Presentation:** 2
**Contribution:** 2
**Rating:** 4
**Confidence:** 2

**Summary:**

This paper introduces a linguistics-based approach for converting English-described mathematical text into predicate logic-based formulas. The proposed pipeline transforms the input text through three main stages: generating a Syntactic Parse Tree, constructing a Standardized Grammar Tree, and producing the final Predicate Logic-based Formula. To evaluate the method, the authors extracted 4,339 sentences from a Calculus textbook, successfully converting 3,380 of them into correct predicate logic-based formulas—achieving an accuracy of 77.90%.

**Strengths:**

- The autoformalization method proposed in this paper is almost purely symbolic, offering a rigorously structured and interpretable alternative to data-driven approaches. It meticulously constructs the Syntactic Parse Tree, Standardized Grammar Tree, and Predicate Logic-based Formula, together with explicit transformation rules between them—forming a coherent and well-engineered symbolic pipeline. In an era dominated by neural autoformalization methods based on large language models, this symbolic approach provides fresh insights and a valuable complementary perspective.
- The proposed method does not rely on LLMs, and thus avoids the need for costly model training or inference. This design greatly reduces computational overhead and improves practical accessibility, making the approach well-suited for resource-limited environments while maintaining transparency and reproducibility.

**Weaknesses:**

- The paper is difficult to read and not reader-friendly. It introduces numerous linguistic concepts, definitions, and transformation rules, but many readers, especially those from the AI community, may not have a systematic background in linguistics. The presentation would benefit from simpler explanations, illustrative examples, and clearer visualizations to make the method more accessible and easier to follow.
- The dataset is limited in scope, as all examples are extracted solely from a Calculus textbook. This narrow focus restricts the diversity of problem types and linguistic structures covered, which may limit the generalizability of the proposed method to other mathematical domains such as algebra, geometry, or discrete mathematics.
- The experimental evaluation is not sufficiently comprehensive. The study only reports the accuracy of formula conversion, without deeper analysis or ablation studies. Additional experiments, e.g., comparisons with LLM-basedl autoformalization approaches, would strengthen the empirical validation and provide a more convincing demonstration of the method’s effectiveness.

**Questions:**

1. The authors only obtained predicate logic–based formulas without translating them into established formal languages such as Lean, Isabelle, or Coq. As a result, it is not possible to make a direct comparison with existing autoformalization methods, which typically operate within those formal frameworks. Do the authors plan to extend their pipeline to support translation into these formal languages?

2. If the pipeline is extended to target a concrete proof assistant, how do the authors expect its performance to compare with current state-of-the-art autoformalization methods (including LLM-based approaches)? In particular, would the symbolic nature of the pipeline give it advantages in precision, while possibly limiting coverage, and how do the authors plan to measure this trade-off?

3. The authors claim that commonly used formal languages lose certain semantic information (e.g., tense, aspect, modality, and various adverbials) that may still be useful for reasoning or inference. However, this argument is not fully substantiated. Could the authors provide specific examples from mathematical texts where these linguistic features meaningfully affect logical interpretation or inference outcomes? For instance, how might differences in tense or modality alter the truth conditions or reasoning structure of a mathematical statement?

---

### Meta-Review · Area_Chair_N83L · 2026-01-01

**Summary:**

This paper introduces a linguistics-based method of using predicate logic-based formulas for autoformalizing English-described mathematical text into formal formulas. Experimental evaluations inlcude accuracy and a case study.

**Reviewer Concerns:**

* Generalizability concern due to the high reliance on Stanford CoreNLP and a single textbook.
* The experiments and evaluation metrics are not sufficiently comprehensive to demonstrate the effectiveness of the method.
* The paper is not well-written enough.

**Reviewer Scores:**

We have no idea how the reviewer will adjust the scores, because the authors did not participate in the rebuttal

---

### Decision · Program_Chairs · 2026-01-26

Reject